# A Theoretical Insight of Histogram Binning and Extending to Multi-label Classification

## Abstract

Learning well-calibrated probabilistic predictions is crucial as neural networks and machine learning models are increasingly employed in critical tasks nowadays. While there exist several post-processing methods aimed at calibrating output probabilities, most lack proper theoretical justification; in other words, they have typically only been validated on limited datasets and models to report empirical results. This work is divided into two parts. In the first part, we analyze some post-processing calibration methods from a geometrical perspective and demonstrate that calibrated outcomes consistently reduce Expected Calibration Error (ECE) while increasing accuracy. In the second part, we present a previously unexplored framework for calibrating the outcomes of multi-label problems by addressing multiple binary calibration problems. To achieve this, we introduce a novel concept of ECE for multi-label problems and provide substantial theoretical rationale for our approach. Experimental results demonstrate the feasibility and efficacy of our method in practice.

## 1 Introduction

Deep learning methods have dramatically improved prediction accuracy. This leads the usage of these models in complex decision making tasks like medical image processing(Litjens et al., 2017; Shen et al., 2017), disease diagnosis(Caruana et al., 2015; Jiang et al., 2012), autonomous driving(Bojarski et al., 2016) etc and Neural network is an essential part of these systems. Though the ability of neural networks has increased in last decade, it is essential for a model to have the ability to indicate when the prediction is incorrect. The confidence for each prediction does this job for us- the higher the confidence the more we can trust the result of the model. For intriguing tasks mentioned above confidence plays a key role. As for example we can think of self-driven cars(Bojarski et al., 2016; Tian et al., 2018) which uses neural network based object detection module to determine whether there is some pedestrian or some blockage in front of the car. If the module cannot predict these with a very high confidence then the car must rely on other sensors for braking. Unfortunately, for many modern deep learning models the algorithms mainly focuses on accuracy rather than producing well calibrated results. In fact it is very surprising that modern neural networks have become less calibrated(Guo et al., 2017) though their accuracy has increased. (Guo et al., 2017) has showed that a 5 layer Le-Net(LeCun et al., 1998) is more calibrated than a 110 layer Res-Net(He et al., 2016). On the other hand Res-Net is more accurate than Le-Net. This also establishes an intuition that accuracy is independent or orthogonal to confidence.

Several post processing calibration methods are used widely to calibrate the outputs of the neural networks and other machine learning models. These methods can be divided mostly into two parts: Parametric and Non-parametric. An example of parametric method is Platt's scaling(Platt et al., 1999) which applies a sigmoidal transformation to predictive probabilities to give more calibrated output. Some Non-parametric method consists Histogram Binning(Zadrozny & Elkan, 2001), Isotonic Regression(Zadrozny & Elkan, 2002) etc. In Histogram Binning the outputs of the Binary Classifier are sorted then they are divided into $B$ numbers of equally spaced bins and a binning parameter($\theta_m$ for bin $B_m$) is assigned to each and every bin. The predictive probability now becomes $\theta_m$ if it falls into the bin $B_m$.

Despite these methods give better calibrated results in a few studies(Naeini et al., 2015; Zadrozny & Elkan, 2001), we pointed out that these results are mostly empirical, based on some very well known data sets and models. In the following sections we will see that Histogram Binning changes the output probability to give us calibrated results. But there is no theoretical background which suggests that calibrating using Histogram Binning will surely give us better calibrated and more accurate results(for any data sets and models). We find this question very important as these methods can be used to calibrate the outcomes of very delicate tasks.

Although traditional neural networks and supervised learning algorithms work successfully for multi-class classification, the real world is much more complicated and the assumption of only one class label associated with each input does not fit well. In most of the real world objects there will be multiple semantics associated with it. To name a few, a single medical image(Bustos et al., 2020) is related to multiple radiographic findings; In text classification(Minaee et al., 2021) one single piece of text might cover different genres like thriller, drama, romance; In self driving cars there will be several different labels like pedestrians, roads, humans are often present in one single frame. To tackle multiple semantics of a single object we resort to multi-label classification methods. Medical Image analysis, Self Driving cars etc are very delicate multi-label classification task for which precise confidence of the classifiers are needed keenly. Unfortunately, there are not much studies with proper theoretical justification regarding calibration of multi-label problems.

This study presents a theoretical rationale of Histogram Binning along with a framework to apply this method to calibrate multi-label classification problems. Our approach to Histogram Binning is from a geometric and functional point of view. Usage of Large Sample Theory endorses the validity of this method. For multi-label classification framework, firstly two metrics are proposed to measure the Total and Average Expected Calibration Error(ECE) as ECE(Naeini et al., 2015) proposed in previous works is not designed to measure the calibration error for Multi-Label classification problems. Then we have proposed a previously unexplored method which combines multiple binary Histogram Binning problems to work for multi-label problems. Theoretical rationale behind our method is also provided along with. The experimental results presented for multi-label problems endorses our claims and verifies that out method works well in practice.

## 2 Definitions and Background

Let $\mathcal{X} = \{X_1, \ldots, X_n\}$ is the set of input data and the set of output labels are $\mathcal{Y} = \{1, 2, \ldots, K\}$. Both of them are random variables following a joint distribution which can be written by using Bayes Rule $f(X, Y) = f(Y|X)f(X)$, where $X \in \mathcal{X}$ and $Y \in \mathcal{Y}$. Assume that $h$ is a Neural Network(NN) such that $h : \mathcal{X} \to \mathbb{R}^K$ and $\sigma_{SM}$ is the softmax function. Define $\sigma_{SM} : \mathbb{R}^K \to \mathcal{P}_K$ where $\mathcal{P}_K = \{(p_1, \ldots, p_K) \in \mathbb{R}_+^K | p_1 + \cdots + p_K = 1\}$ and $\hat{y}(X_i) = \arg\max_{1 \leq k \leq K}\{\sigma_{SM}(h(X_i))_k\}$. An example of perfect calibration can be seen in a weather prediction model forecasting 80% chance of rain next day consistently throughout the year, resulting in rain occurring exactly 80% of the days during the time period. In short, whenever predicted class is $\hat{Y}$ and predicted probability is $\hat{P}$ for a neural network $h$, perfect calibration implies $P_h(\hat{Y} = y|\hat{P} = p) = p$. As we can see from the definition, with finite number of sample points it is impossible to calculate so we resort to empirical approaches.

### 2.1 Expected Calibration Error

Let us define $\hat{y}_1, \ldots, \hat{y}_n$ as the predicted labels and $\hat{p}_1, \ldots, \hat{p}_n$ be the predictive probabilities of a binary classification problem. $\hat{p}_i$'s of such NNs is the confidence of $i^{th}$ instance to belong to the positive class. We will assume that these are sorted in ascending order, that means $\hat{p}_1 \leq \ldots \leq \hat{p}_n$. The total number of bins in between $[0, 1]$ is M. If there are $n_m$ number of output probabilities inside the bin number $m$ then $n = \sum_{j=1}^{M} n_j$. For bin $m$, the predicted probabilities are defined by $\hat{p}_m^k$ and the corresponding response by $\hat{y}_m^k$ for $k = 1, 2, \ldots n_m$. The responses are independent of each other. Under the assumption of null hypothesis which is perfect calibration, we can say that $\hat{y}_m^k$ is the outcome of a Bernoulli trial with an

expected value of $\hat{p}_m^k$. We sort the predicted probabilities such that $\hat{p}_m^i \leq \hat{p}_m^j$ when $i \leq j$ and $\hat{p}_{m_1}^l \leq \hat{p}_{m_2}^l$ when $m_1 < m_2$. For convenience we define $\hat{p}_0^1 = 0$; $\hat{p}_{m+1}^1 = 1$.

To define the confidence of a bin $B_m$, we take the average of all predicted probabilities lying inside that bin. Then we also define the fraction of positive instances($pos_m$) and the expected accuracy of the $m^{th}$ bin.

$$conf_m = \frac{1}{n_m} \sum_{i=1}^{n_m} p_m^i$$

$$pos_m = \frac{1}{n_m} \sum_{i=1}^{n_m} \mathbb{1}(\hat{y}_m^i = 1)$$

and

$$acc_m = \frac{1}{n_m} \sum_{i=1}^{n_m} \mathbb{1}(\hat{y}_m^i = y_m^i)$$

Expected Calibration Error(ECE) metric is commonly used to measure the calibration error, first proposed by (Naeini et al., 2015) to calibrate binary classification problems. ECE measures the discrepancy between the fraction of positive instances and confidence of a model thus captures the miscalibration. To compute this measure predicted probabilities are sorted into $M$ equally spaced bins, $B_m$, each of size $\frac{1}{M}$. The empirical estimate can be written as:

$$ECE = \sum_{m=1}^{M} \frac{n_m}{n} |pos_m - conf_m|. \tag{1}$$

## 3 Trustworthiness of calibration

We are going to use calibrated results in very critical tasks like medical image classifcation, disease detection, self driving cars etc. as a well calibrated NNs should reflect the true confidence of the prediction being correct. But till now only a few data sets were used to validate the methods of calibration. In real-world scenarios where data sets can be intricate, it's pertinent to question whether calibration methods consistently yield improved calibration while maintaining overall accuracy, or if they might occasionally compromise the performance. In this work we will analyze a model agnostic method Histogram Binning(HB) and show analytically with some minor assumptions that HB does not compromise the performance of NNs.

### 3.1 Histogram Binning

Histogram Binning(Zadrozny & Elkan, 2001) uses bins to calibrate the outcomes of a NN. Probabilities predicted by NN, $\hat{p}_i$s are sorted and divided into $M$ mutually exclusive bins namely $B_1, \ldots, B_M$. Each bin is assigned a score say $\theta_m$ for bin $m$. Whenever the predicted probability $\hat{p}_{te}$ for a test input falls inside the bin $B_m$, the calibrated probability $\hat{q}_{te}$ takes the value $\theta_m$. The bin edges can be defined as $0 = a_1 < \ldots < a_{M+1} = 1$ hence $B_m = (a_m, a_{m+1}]$. The bins can be chosen such that they are equally spaced or there are equal numbers of points inside each bin. In this work we have considered equally spaced bins. As the bins are equally spaced, the $m^{th}$ bin is going to be $B_m = (\frac{m-1}{M}, \frac{m}{M}]$. To choose the scores for every bin we will resort to the following equation:

$$(\theta_1, \ldots, \theta_M) = \arg \min_{\theta_1, \ldots, \theta_M} \sum_{m=1}^{M} \sum_{i=1}^{n} \mathbb{1}(a_m \leq \hat{p}_i \leq a_{m+1})(\theta_m - y_i)^2 \tag{2}$$

where $y_i \in \{0, 1\}$ is the ground truth of $i^{th}$ input.

Below we will state and prove a theorem in which we will see association of HB with the scores of each bin.

**Theorem 1.** *The binning scores are the ratio of number of positive classes and the total number of instances in the bin. In short, $\theta_m = \frac{n_m^{(1)}}{n_m}$ where $n_m^{(1)}$ is the number of positive instances inside the bin $m$.*

*Proof.* The equation 2 is sum of squares so the right hand side cannot be negative. So we have

$$\sum_{m=1}^{M} \sum_{i=1}^{n} \mathbb{1}(a_m \leq \hat{p}_i \leq a_{m+1})(\theta_m - y_i)^2 = 0$$

By differentiation we get

$$\sum_{i=1}^{n_m} (\theta_m - y_i) = 0 \tag{3}$$

$y_i = 0$ or 1. From equation 3 it can be inferred that $\theta_m = \frac{n_m^{(1)}}{n_m}$.

$\square$

From this theorem we get to see that the scores actually corresponds to the amount of positive instances in a particular bin. But it is still questionable whether these calibrated scores are actually accurate or not or whether we can trust these new predicted probabilities over the uncalibrated outputs produced by neural networks because the results in previous studies are all empirical. These issues are not yet addressed analytically and hinders a user to use the calibrated probabilities produced by HB safely. In the next lemma and theorem we will see that HB always produces outputs which are better calibrated and the accuracy is also maintained.

**Lemma 1.** *Let $(\mathcal{X}, \mathcal{Y})$ be a binary classification dataset, where $Y_i \in \{0, 1\} \ \forall \ \ Y_i \in \mathcal{Y}$ and a neural network $h : \mathcal{X} \to \mathbb{R}$. Let $\sigma_{SG}$ be a sigmoid function and the interval $[0, 1]$ is divided into $M$ equally spaced bins; $I_m$ be the $m^{th}$ bin such that $B_m = (\frac{m-1}{M}, \frac{m}{M}]$. Assume that $h$ is injective function. Let $C_m$ be the cluster centre of all points $X_i \in \mathcal{X}$ such that $\sigma_{SG}(h(X_i)) \in B_m$. Then we can say that for new instances $X$, $X'$ if $\sigma_{SG}(h(X)) \in B_m$ and $\sigma_{SG}(h(X')) \in B_j$; then $d(C_m, X) < d(C_m, X')$ for all $j \in \{1, 2, \ldots, m-1, m+1, \ldots, M\}$ where $d$ is the distance metric.*

*Proof.* $h$ is an injective(if we use any non-polynomial analytic activation function then we have that neural network is injective and some other conditions of injectivity is described in (Puthawala et al., 2022)) function, thus $\sigma_{SG} \circ h : \mathcal{X} \to [0, 1]$ is injective too. Let $f_{nn} = \sigma_{SG} \circ h$. For any new instance $X$, $f_{nn}(X) \in B_m \implies X \in f_{nn}^{-1}(B_m)$; $B_m$ is a convex set. We need to show that $f_{nn}^{-1}(B_m)$ is a convex set.

Let, $\mathbf{x}, \mathbf{y} \in f_{nn}^{-1}(B_m)$. Then $\alpha f_{nn}(\mathbf{x}) + (1 - \alpha) f_{nn}(\mathbf{y}) \in B_m$ for any $\alpha \in [0, 1]$. So $f_{nn}^{-1}(\alpha f_{nn}(\mathbf{x}) + (1 - \alpha) f_{nn}(\mathbf{y})) \in f_{nn}^{-1}(B_m)$. As $f_{nn}$ is injective and continuous so by the property of inverse mapping $\alpha \mathbf{x} + (1 - \alpha) \mathbf{y} \in f_{nn}^{-1}(B_m)$. So, $S_m = f_{nn}^{-1}(B_m)$ and $S_j = f_{nn}^{-1}(B_j)$ are disjoint convex sets.

As $S_m$ and $S_j$ are disjoint convex sets, by Hyperplane Separation Theorem there exists a non-zero vector $v$ and $c \in \mathbb{R}$ such that $\langle x, v \rangle \geq c$ and $\langle y, v \rangle \leq c$ for all $x \in S_m$ and $y \in S_j$. Now, $X' \in S_j$ which is outside of $S_m$ and between $S_m$ and $S_j$ there is hyperplane that is separating these two. So $d(C_m, X) < d(C_m, X')$. $\square$

The consequence of the lemma 1 is that the inputs for which the neural network produces the output probability within same interval are tend to be similar than the points for which the output probability lies outside the specific interval. So, for any input if the output probability lies inside the said interval then the points are expected to have properties more similar to the cluster centre of inverse image of that interval than the cluster centre of the inverse image of the other intervals.

**Theorem 2.** *A calibrated probability obtained by solving the equation 2 is always going to be more well calibrated and accurate compared to uncalibrated outputs.*

*Proof.* $X$ be a input and $f_{nn}(X) \in B_m$. We have $\theta_m = \frac{n_m^{(1)}}{n_m}$. $\lim_{n_m \to \infty} \theta_m = \Theta_m$ by the Law of Large Numbers. $\Theta_m$ is representing the true fraction of samples to be of positive class for very large $n_m$. So for any $X \in f_{nn}^{-1}(B_m)$, randomly chosen(this also represent any element used as a input of the neural network) the output probability ideally should be $\Theta_m$. In this way it is actually representing the true probability of being an element belong to the positive class. So, for bin $I_m$; $|pos_m - conf_m| > |pos_m - \Theta_m|$ making it better calibrated.

If $X_{te} \in f_{nn}^{-1}(B_m)$ then $X_{te}$ will be assigned to the positive class with probability $\theta_m$. By lemma 2 $d(X_{te}, C_m) < d(X_{te}, C_j)$ implies that $X_{te}$ will be more similar to the points in $f_{nn}^{-1}(B_m)$. In case of wrongly classified points to the zero(positive) class but pretty high(low) $\theta_m$ suggests that for most of the data points in $f_{nn}^{-1}(B_m)$ the true class will be the positive(zero) class. So the accuracy will increase too. $\square$

## 4 Calibration for Multi-class Classification(MCC)

Now we will present a version of the histogram binning method for multi-class classification by mainly following (Zadrozny & Elkan, 2002). Let $h : \mathcal{X} \to \mathbb{R}^K$ be a neural network where $K$ is the number of classes. $\sigma_{SM}$ be the softmax function as defined previously. As the output of every class label will be in between $[0, 1]$, we will divide the outputs of every class label into $M$ equally spaces bins. The binning parameter of the $m^{th}$ bin of the $i^{th}$ class is $\theta_m^{(i)}$. If the output probability of $i^{th}$ class of the $j^{th}$ input, $p_j^{(i)}$, falls inside the bin $B_m^{(i)}$ then calibrated probability $q_j^{(i)} = \theta_m^{(i)}$. The calibrated output probability is going to be $\{q_j^{(1)}, \ldots, q_j^{(K)}\}$. But $\sum_{l=1}^{K} q_j^{(l)} \neq 1$ in general. So, after normalization the output probability is going to be $\{\frac{q_j^{(1)}}{\sum_{l=1}^{K} q_j^{(l)}}, \ldots, \frac{q_j^{(K)}}{\sum_{l=1}^{K} q_j^{(l)}}\}$ and the confidence of the correct class is going to be $max\{\frac{q_j^{(1)}}{\sum_{l=1}^{K} q_j^{(l)}}, \ldots, \frac{q_j^{(K)}}{\sum_{l=1}^{K} q_j^{(l)}}\}$.

## 5 Calibration for Multi-label Classification(MLC) Problems

Multi-label classification is a machine learning task where a classifier learns to associate multiple labels to an output instead of a single one. For label set $\widetilde{\mathcal{Y}} = \{1, \ldots, K\}$ and an input $\widetilde{X} \in \widetilde{\mathcal{X}}$, the classifier tries to predict $\tilde{\mathbf{y}} \subseteq \widetilde{\mathcal{Y}}$. $\tilde{\mathbf{y}}$ can be any subset of $\widetilde{\mathcal{Y}}$ in case of multi-label classification.

Let $\tilde{h}$ be a neural network for multi-label classification such that $\tilde{h} : \widetilde{\mathcal{X}} \to \mathbb{R}^K$ and $\sigma_{SG}$ be the sigmoid function. Now, $\tilde{h} = (\tilde{h}_1, \ldots, \tilde{h}_K)$ where $\tilde{h}_1 : \widetilde{\mathcal{X}} \to \mathbb{R}$ and $\tilde{p}(\widetilde{X}) = \{\tilde{p}_1(\widetilde{X}), \ldots, \tilde{p}_K(\widetilde{X})\}$ where $\tilde{p}_l(\widetilde{X}) = \sigma_{SG}(\tilde{h}_l(\widetilde{X}))$. If $\hat{\tilde{\mathbf{y}}} \subseteq \widetilde{\mathcal{Y}}$ is the predicted set of labels for the input $\widetilde{X}$ then $l \in \hat{\tilde{\mathbf{y}}}$ if $\tilde{h}_l(\widetilde{X}) > 0$. In this case of multi-label classification, each of $\tilde{h}_l(\widetilde{X})$ can be thought of as an individual neural network involved in predicting whether a particular label $l$ is in the input. In other words, Multi-label classification consists of multiple binary classifiers and the predicted label is going to be a multi-hot encoded vector instead of a one-hot-encoded vector in case of multi-class classification.

### 5.1 Calibrating the Probabilities

In multi-label classification, predicting each label is a task of one binary classifier. So we will calibrate each of these classifiers separately and define two types of ECE. We will produce first results of calibrating a multi-label classifier in this way. Also we will provide a theoretical insight and produce the first theoretical justification for calibrating the multi-label classification.

## 5.2 Calibrating with Histogram Binning

We will treat the multi-label problem as multiple binary classification problem. For this we need to divide each binary output to $M$ equally spaced bins. Let $(\widetilde{X}_j, \widetilde{Y}_j)_{j=1}^n$ be data points and

$$\mathscr{Y}_i = (\mathscr{Y}_i^{(1)}, \ldots, \mathscr{Y}_i^{(K)}); \qquad \mathscr{Y}_i^{(l)} = \begin{cases} 1, & \text{if } l \in \widetilde{Y}_i \\ 0, & \text{o.w.} \end{cases}$$

To calibrate the NNs for multi-label classification problems we will calibrate each $\tilde{h}_l$, $l \in \{1, \ldots, K\}$ separately. We assume that all labels are independent so calibrating them in this way will not cause any loss of information. The uncalibrated predicted probabilities related to $\tilde{h}_l$, $\hat{p}_i^{(l)}$, are divided into $M_l$ mutually exclusive bins $\{B_1^{(l)}, \ldots, B_{M_l}^{(l)}\}$. The bin edges for label $l$ is defined as $0 = a_1^{(l)} < a_2^{(l)} < \ldots < a_{M_l+1}^{(l)} = 1$. The bin edges can be chosen different for different labels but in this work we have considered equal length and equal number of bins for all labels. For each bin in each label, $B_m^{(l)}$, a binning score $\theta_m^{(l)}$ is assigned. For a new prediction $\hat{p}_{te}^{(l)} \in B_m^{(l)}$, the calibrated outcome $\hat{q}_{te}^{(l)} = \theta_m^{(l)}$. $\theta_m^{(l)}$ is chosen to minimize the following equation:

$$(\theta_1^{(l)}, \ldots, \theta_M^{(l)}) = \arg \min_{\theta_1^{(l)}, \ldots \theta_M^{(l)}} \sum_{m=1}^M \sum_{i=1}^n \mathbb{1}(a_m^{(l)} \leq \hat{p}_i^{(l)} \leq a_{m+1}^{(l)})(\theta_m^{(l)} - \mathscr{Y}_i^{(l)})^2 \tag{4}$$

We found that the ECE metric in equation1 that is orginally defined by (Naeini et al., 2015) is only for calibrating binary classification problems; not suitable for multi-label problems. Hence, we define the two measures of ECE for multi-label problems calculating total and weighted average of calibration error denoted by $ECE_{tot}$ and $ECE_{wavg}$ respectively. Mathematically they are defined as follows:

$$ECE_{tot} = \sum_{l=1}^K \sum_{m=1}^M \frac{n_{m_l}}{n} |pos_m^{(l)} - confidence_m^{(l)}| \tag{5}$$

$$ECE_{wavg} = \frac{1}{n} \sum_{l=1}^K \left( (\sum_{i=1}^n \mathscr{Y}_i^{(l)}) \cdot (\sum_{m=1}^M \frac{n_{m_l}}{n} |pos_m^{(l)} - confidence_m^{(l)}|) \right) \tag{6}$$

$pos_m^{(l)}$ and $confidence_m^{(l)}$ is the number of positive instances and confidence of $l^{th}$ label for which the predictive probabilities falls under $m^{th}$ bin. Equation 5 describes the total ECE of all labels. This notion is useful as we have used binary relevance in our multi-label classification problem and the labels are mutually independent. In equation 6 we have chosen weighted average instead of simple average because there might be a scarcity of particular label in the inputs. This kind of setting is quite common in various real world situations. If there are not much presence of sample for a label in the inputs, calibrating that particular label might cause a big change in calibration which from our point of view does not reflect the truth. This might be more problematic if there are very less number of labels present(say 3-5). Weighted average of ECE mitigates the problem by multiplying weights(fractions of each label present in the input) to every label.

**Theorem 3.** *The binning score for $m^{th}$ bin at label $l$, $\theta_m^l$, is going to be $\frac{n_{m_l}^{(1)}}{n_{m_l}}$, where $n_{m_l}^{(1)} = |D_{m_l}^{(1)}|$; $D_{m_l}^{(1)} = \{\widetilde{Y}_i | l \in \widetilde{Y}_i \text{ and } a_m \leq \hat{p}_i^l \leq a_{m+1}\}$ and $n_{m_l} = |D_{m_l}|$; $D_{m_l} = \{\widetilde{Y}_i | a_m \leq \hat{p}_i^{(l)} \leq a_{m+1}\}$.*

We are going to present a generalized version of lemma 1 for the multi-label classification. First we will prove it for each individual binary classifier then we will extend it to the multi-label problems.

**Lemma 2.** *Let $(\widetilde{\mathcal{X}}, \mathscr{Y})$ be a multi-label dataset where $\mathscr{Y}_i \in \{0,1\}^K \ \forall \mathscr{Y}_i \in \mathscr{Y}$. The interval $[0,1]$ is divided into $m$ equally spaced bins $B_m^{(l)} = (\frac{m-1}{M}, \frac{m}{M}]$ for each label $l$. Assume that $\tilde{h} : \widetilde{\mathcal{X}} \to \mathbb{R}^K$ is a neural network which is injective. Let $C_m^{(l)}$ be the cluster centre of all points $\widetilde{X}_i \in \widetilde{\mathcal{X}}$ such that $\sigma_{SG}(\tilde{h}_l(\widetilde{X}_i)) \in B_m^{(l)}$. It can be said that for any new instances $\widetilde{X}, \widetilde{X'}$ if $\sigma_{SG}(\tilde{h}_l(\widetilde{X})) \in B_m^{(l)}$ and $\sigma_{SG}(\tilde{h}_l(\widetilde{X'})) \in B_j^{(l)}$ respectively, then $d(C_m^{(l)}, \widetilde{X}) < d(C_m^{(l)}, \widetilde{X'})$ for any $j \in [M] \setminus m$ where $d$ is a distance metric.*

*Proof.* Let $f_{nn}^{(l)} = \sigma_{SG} \circ \tilde{h}_l$. Hence $f_{nn}^{(l)}$ is a continuous and injective function. $B_m^{(l)}$ is a convex set. By the properties of continuous and injective function, $S_m^{(l)} = f_{nn}^{(l)^{-1}}(B_m^{(l)})$ and $S_j^{(l)} = f_{nn}^{(l)^{-1}}(B_j^{(l)})$ are disjoint convex sets. $C_m^{(l)}$ be the cluster centre of the set $S_m^l$

As the sets are convex and disjoint, again by Hyperplane Separation Theorem, the existence of a hyperplane between $S_m^{(l)}$ and $S_j^{(l)}$ guarantees that $d(C_m^{(l)}, \widetilde{X}) < d(C_j^{(l)}, \widetilde{X}')$. □

With this lemma we are now ready to show the validity of our extension theoretically. Experimental results will be provided to endorse our claims.

**Theorem 4.** *For any given neural network and training set, the calibrated probabilities of label 'l' obtained by solving the equation 4 is going to be better calibrated and more or equally accurate compared to an uncalibrated one.*

*Proof.* Let $\widetilde{X}$ be input to the model and $f_{nn}(\widetilde{X} \in B_m^{(l)}$. By the Law of Large numbers $\theta_m^{(l)} \xrightarrow{n_m \to \infty} \Theta_m^{(l)}$. $\Theta_m^{(l)}$ is representing the true fraction of inputs that belongs to the positive class for the interval $I_m^{(l)}$ for very large $n_{m_l}$ ideally. For any $\mathcal{X} \in f_{nn}^{-1}(B_m^{(l)})$, the output probability should ideally be $\Theta_m^l$. Hence, for bin $B_m^{(l)}$; $|pos_m^{(l)} - \Theta_m^{(l)}| < |pos_m^{(l)} - confidence_m^{(l)}|$. By using the ineqality in 7 and 8 we have proved our claim.

Let $X_{te} \in f_{nn}^{-1}(B_m^{(l)})$ then Lemma 2 implies that $X_{te}$ will be more similar to the points in $B_m^{(l)}$. In case of wrongly classified points to zero class and very high $\theta_m^{(l)}$ the correct class is more probable to be actually positive which indicates the increase in accuracy.

□

So we have established some genralized results which guarantees the effectiveness of Histogram Binning for both multi-label and multi-class problems. Theorem 4 is an extended theorem of calibration in case of multi-label classification problems. We have proved that under some conditions the calibrated probabilities obtained by using Histogram Binning are going to give us better estimates. Also this result makes the calibrated estimates trustworthy.

## 6 Experiments and Results

This section describes some experiments that we have performed to check the validation of our theory in case of Multi-label image and text datasets. We have used $ECE_{tot}$ and $ECE_{wavg}$ for multi-label calibration as a measure to check the calibration error. To check the accuracy we have used Hamming Loss(HL) which is described below:

**Hamming Loss:** If $\hat{\mathscr{Y}}_i$ and $\mathscr{Y}_i$ be the predicted labels and true labels respectively for the $i^{th}$ input, then the hamming loss is defined as:

$$HL = \frac{1}{NL} \sum_{l=1}^{N} \sum_{i=1}^{N} \hat{\mathscr{Y}}_i^{(l)} \oplus \mathscr{Y}_i^{(l)} \tag{7}$$

### Datasets

We have used two image datasets and one text dataset mainly for our experiments. The first one is a dataset(shr) from Kaggle, publicly available for multi-label classification problems. There are 7843 images in total and each image has one or more than one label among 10 labels. The dataset is originally developed to find the personality of people based on the pictures they share in facebook and other social media. Another is the famous PASCAL-VOC(Everingham et al.) dataset which consista several images from flicker mainly. It has in total 20 labels and each and every image contains one or more number of labels. The third one is

| Dataset and | Uncalibrated | | Calibrated | |
|---|---|---|---|---|
| Model | $ECE_{wavg}$ | $ECE_{tot}$ | $ECE_{wavg}$ | $ECE_{tot}$ |
| Kaggle Image Dataset; ResNet 50 | 0.0406 ±0.0052 | 0.20 ±0.02 | **0.014** ±0.0024 | **0.12** ±0.008 |
| PASCAL-VOC; ResNet 50 | 0.03819 ±0.0048 | 0.2851 ±0.016 | **.22** ±0.0169 | **.207** ±0.017 |
| Propaganda PRC; BERT | 0.037 ±0.0052 | 0.202 ±0.07 | **0.0214** ±0.00152 | **0.11** ±0.009 |

Table 1: $ECE_{wavg}$ and $ECE_{tot}$ for $M = 10$ bins for kaggle image dataset, PASCAL-VOC dataset and propaganda PRC dataset before and after calibration

Dataset of Propaganda Techniques of the State-Sponsored Information Operation of the People's Republic of China(PTSI-PRC). This dataset consists 9950 tweets in Mandarin with 20 different propaganda techniques. We have used several randomized partitions for each of the dataset in our experiments to get the variability of calibration measures.

To conduct experiments on image datasets we have used ResNet50 CNN, a variant of ResNet with 50 layers. As it has high performance in ImageNet competition, we decided to choose this network as our backbone in our experiments. We have tweaked the model a bit to use it in multi-label classification. The softmax layer is removed and instead of that sigmoid function is used to represent probability of presence of each label in an input. Then the output is calibrated with the calibrating functions and ECE is measured for both uncalibrarted and calibrated datasets. This procedure is followed in all of our experiments which contains Images. To conduct our experiment in text datasets we have used BERT(Devlin et al., 2018) model as in (Chang et al., 2021), an open source framework for different tasks in NLP. BERT enhances machine understanding of ambiguous language by leveraging surrounding texts. All of the calibration related results are presented in the table 1. The code for PASCAL-VOC can be found here: `https://github.com/drunksailors/multi-label-calibration`

It can be seen that our experimental result is on par with our theoretical results. For all three datasets that are presented here, both $ECE_{wavg}$ and $ECE_{tot}$ are decreasing significantly after calibration. Our method works for text data as well. In theorem 4 we have claimed that the accuracy will improve along with better calibrated results. To endorse our claim of betterment of accuracy, we used the same datasets and calculated the Hamming Loss to measure the discrepancy between predicted and actual labels before and after calibration. The results in table 2 establishes out claim. So, we have devised a method for calibration of Multi-label problems which not only increases the calibration also does not hamper the accuracy.

# 7 Conclusion

In this work we have provided theoretical insight of using Histogram Binning. We have proved that Histogram Binning will always give us better calibration and more accurate results rather than an uncalibrated one. Also a framework for calibrating multi-label classification problems is provided along with theoretical rationale. We have presented experimental results which endorses that our theoretical formulation will work in practice.

In this work the bin numbers are fixed at $M = 10$. One future direction of research can be to be able to choose the optimal number of bins and types of bins(equally spaced or equal frequency or any other type). Bayesian Binning Quantile(Naeini et al., 2015) can be extended to calibrate multi-label problems. Also some train time methods(Hebbalaguppe et al., 2022; Müller et al., 2019) can be combined with the model agnostic post-hoc methods such as ours to improve the calibration further more. Our assumption was neural networks

| Dataset and model | Uncalibrated | Calibrated |
|---|---|---|
| Kaggle Image Dataset; ResNet 50 | 0.0379 | **0.03552** |
| PASCAL-VOC; ResNet 50 | 0.0711 | **0.07** |
| Propaganda PRC; ResNet 50 | 0.0242 | **0.0229** |

Table 2: Hamming Loss before and after calibration

are injective in out theoretical results. In future one can try to exclude the condition of injecticvity and try to prove more general results by showing the convergence of binning scores.

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
