# OpenReview forum: "A Theoretical Insight of Histogram Binning and Extending to Multi-label Classification"
_TMLR — Rejected by TMLR_

### Review · Reviewer_s1Tb · 2024-07-26

**Summary Of Contributions:**

This paper contributes to the understanding of calibration in neural networks by providing a theoretical foundation for why histogram binning consistently improves the calibration and accuracy of neural network outputs, and extends it to the multi-label setting.

**Audience:**

Yes

**Claims And Evidence:**

Yes

**Requested Changes:**

1. For writing and formatting:

(1). Those three equations `conf_m`, `pos_m`, and `acc_m` take unnecessary a lot og space in section 2.1. Please format them more compactly.

(2). Some words in the middle of sentences are inconsistently capitalized. For example, sometimes it is `self driving cars`  and other times it is `Self Driving cars` . Please ensure consistent capitalization throughout the paper.

(3) There is inconsistency in referring to equations. `equation`  and `Equation` are used interchangeably. Please standardize this.

(4) The grammar is very poor, with past and present tenses mixed together.

(5) `consista` should be `consists`.

(6) Review and adjust the number of decimal places used in the tables to ensure consistency. For example, if four decimal places are used in one instance (e.g., 0.0711), then four decimal places should be used consistently throughout the table (e.g., 0.0700 after calibration).

2. It would be interesting to use more recent and large datasets.

3. Some baseline methods should be included for comparison.

4. The author should reason and provide analysis on how ResNet 50 and BERT meet those conditions to be injective.

**Strengths And Weaknesses:**

Strengths:

1. This work provides valuable theoretical backing for an empirically successful calibration method and extends it in a principled way to the multi-label setting.

2. The paper provides a rigorous theoretical analysis of why histogram binning improves calibration and accuracy.

3. They prove and experimentally show that their method for multi-label classification of treating each label independently and calibrating it with histogram binning leads to improved calibration while maintaining accuracy.

Weaknesses:
1. The writing and formatting are poor, please see the `Request Changes`.

2. In the experimental setting, BERT is mentioned as the backbone for the text dataset, as reflected in the results in Table 1. However, in Table 2, the backbone for the text dataset is listed as ResNet50. Is there a reason for this change, or if it is a typographical error?

3. For the Kaggle Image and Propaganda PRC datasets, the sample numbers are mentioned in detail. However, for the PASCAL-VOC dataset, the paper only states 'several images from Flickr' without providing specific details. What is the reason for not describing the PASCAL-VOC dataset in detail? How many samples were exactly used in the experiment?

4. All three datasets in the experiment are too dated and small.

5. No baseline methods for comparison.

6. The number of decimal places in the table results is inconsistent. In Table 2, the results on PASCAL-VOC before calibration is 0.0711, and after calibration is 0.07. It makes me wonder if it is because after calibration the performance becomes worse then the author only kept it as 0.07.

7. There is a weird result in Table 1. For PASCAL-VOC, the ECE_wavg before calibration is 0.03819, and the rest two results on Kaggle and text dataset are also around 0.04 and 0.03. However, The ECE_wavg on PASCAL-VOC after calibration increases to 0.22, but the ECE_tot decreases as expected. This result does not make sense.

8. The assumption that` neural networks are injective` is too strict for many practical deep learning models, it is difficult to hold in practice. According to the referred work ‘Globally Injective ReLU Networks’ in this paper for injective property, ResNet 50 and BERT can hardly meet those conditions.

---

> ### Author Response · Authors · 2024-09-07
>
> Thank you for your thoughtful suggestions. The responses to the weaknesses are provided in the same order in which they were raised.
>
> $\textbf{Answers to Weaknesses}$
>
> 1. Thank you for your feedback regarding the formatting of our writing. We apologise for any confusion caused by the presentation of our work. To address this, we will revise the manuscript to improve its readability and coherence.
>
> 2. Thank you very much for pointing out this error. This is a typographical error. We have used BERT for a text dataset in our experiments. We will revise the manuscript to correct this.
>
> 3. We appreciate the importance of providing clear and specific details about the datasets used in our experiments. We did not provide an in-depth description of the PASCAL-VOC dataset in the paper because it is a widely used and well-known dataset in the computer vision community, with comprehensive documentation available publicly. Given its popularity and the established familiarity within the field, we focused on providing more detailed descriptions for the less familiar datasets such as the Kaggle Image and Propaganda PRC datasets. In our experiments, we used 17125 samples from the PASCAL-VOC dataset to train and validate our model. In the test dataset there are 5183 samples. The training, validation and testing partitions are taken as defined in the paper[a]. We recognize that providing this specific information is important for clarity, and we will include these details in the revised version of the paper.
>
>            [a]Everingham, M., et al. "The PASCAL visual object classes challenge 2012 (VOC2012) results. 2012 http://www. pascal-network. org/challenges." VOC/voc2012/workshop/index. html. 2012.
>
> 4. We chose the PASCAL-VOC dataset because it remains one of the most important and widely-used datasets for multi-label classification in computer vision. Its continued use in many benchmarks and research studies attests to its significance. The Kaggle Image dataset is a newer dataset, providing up-to-date challenges in multi-label classification. The Propaganda PRC dataset, on the other hand, introduces diversity by incorporating a text-based multi-label classification task, expanding the evaluation beyond just image data. Our choice of datasets was deliberately made to ensure diversity, covering both image and text domains. This variety helps demonstrate the generalizability and robustness of our calibration method across different types of data and tasks. These datasets are still used in many important tasks today, making them relevant benchmarks for evaluating multi-label classification methods. However, we would be grateful if you kindly suggest to us some modern dataset with larger sample size . Expanding the range of datasets would certainly strengthen the generalizability of our results, and we would welcome the opportunity to explore newer datasets in future work.
>
> 5. In our experiments, we used multi-label binary cross-entropy (BCE) as the uncalibrated baseline method for comparison. This baseline was used to measure the performance of the proposed post-hoc calibration method. However we appreciate that this was not clearly mentioned in the paper and are willing to revise the manuscript in this regard.
>
> 6. We kept it 0.07 because the original value was 0.07004725. As we have considered values up to 4 decimal points the Hamming Loss(HL) after calibration was 0.07.
>
> 7. The reported value of ECE_wavg (0.22) for the PASCAL-VOC dataset after calibration was indeed a typographical error. The correct value should be 0.022, which aligns with the expected decrease in the ECE_wavg after calibration, consistent with the trends observed in the other datasets. We will correct this error in the revised version of the paper and ensure that all results are accurately reported. Thank you again for bringing this to our attention.
>
> 8. Thank you for pointing out this concern. We assumed that the weight matrices of the Neural Network structure that we are using is of full rank. Then according to definition 1, theorem 2 and theorem 3 of paper [b] the ReLU layer is layerwise injective. As the ReLU layers are linear so their composition is also linear. However, we have found another much easier approach to prove this theorem and we will include it in the revised version of the paper.
>
>            [b] Puthawala, Michael, et al. "Globally injective relu networks." Journal of Machine Learning Research 23.105 (2022): 1-55.
>
> We will also include all  the requested changes that you have mentioned. Thank you for pointing out these discrepancies in our work.

---

### Review · Reviewer_82Zt · 2024-07-29

**Summary Of Contributions:**

This paper explores the success of histogram binning techniques in model calibration. Specifically, the authors provide some theoretical evidence that histogram binning consistently reduce the expected calibration error while increasing the model accuracy. Besides, they generalize the histogram binning to the multi-label classification problems with theoretical analysis. Some evaluation of calibration on several multi-label classification datasets has shown the effectiveness of histogram binning.

**Audience:**

Yes

**Broader Impact Concerns:**

None.

**Claims And Evidence:**

Yes

**Requested Changes:**

1. Please justify how this work motivates further research in model calibration.
2. Please discuss the generalization of the analysis to other post-processing techniques.
3. Please discuss and compare with other model calibration methods for multi-label problems.
4. Please provide more experiments on different models and ablation studies.

**Strengths And Weaknesses:**

Pros:
1. The analysis of histogram binning provides some theoretical justification to the success of histogram binning.
2. The extended analysis of histogram binning in multi-label classification problems shows its effectiveness.

Cons:
1. It is difficult to see how the provided theoretical evidence provides more insights to further research in the field of model calibration.
2. The discussed analysis heavily relies on the existing histogram binning techniques, however, many other post-processing methods are not included in the analysis, such as temperature scaling.
3. Some relevant works of model calibration in the field of multi-label problems are not discussed and compared, such as [a].
4. The experiments are insufficient. More experiments on various models, such as ViTs, should be included. Also, there is no ablation study of histogram binning, such as the number of bins.

[a]. Towards Calibrated Multi-label Deep Neural Networks. CVPR 2024.

---

> ### Author Response · Authors · 2024-09-07
>
> Thank you for your suggestions. The responses to the cons are presented in the same order in which they were raised.
>
>
> $\textbf{Answers to Cons}:$
> 1.  Thank you for your valuable feedback. We appreciate the importance of clearly articulating how our work can advance research in model calibration.
>
>      Our work focuses on providing rigorous theoretical justifications for the use of histogram binning as a calibration method. Specifically, we have proven that histogram binning always provides more accuracy and better calibration. These proofs are significant because they establish a formal foundation for understanding why histogram binning produces superior calibration results. With these theoretical results, our work helps to bridge the gap between empirical performance and theoretical understanding.
>
>        These theoretical insights have important implications for further research in model calibration. Firstly, they can guide the development of new calibration methods that build on the strengths of histogram binning, potentially leading to even more effective approaches. Secondly, our results can serve as a benchmark for evaluating other calibration methods, encouraging researchers to explore alternative techniques that either complement or improve upon histogram binning. Lastly, the formal guarantees we provide may inspire further theoretical investigations into the calibration properties of other methods, contributing to a deeper understanding of the field.
>
>
> 2. You’ve raised a very important point. In our research, we focus on non-parametric calibration methods, particularly histogram binning, and adapt it to multi-label classification problems. Non-parametric approaches like histogram binning are advantageous because they don’t make assumptions about the underlying distribution of predictions, allowing for greater flexibility in estimating the underlying distribution. In contrast, parametric methods like temperature scaling lack this flexibility. Additionally, temperature scaling tends to reduce the confidence of all predicted probabilities, including those that are correct, in order to minimise calibration error. We believe this approach is not ideal, as the goal should be to increase the confidence in correct predictions while reducing the confidence in incorrect ones. We will modify the introduction to add such a discussion.
>
> 3. Thank you for highlighting this recent work. We appreciate the relevance of [a] to the field of model calibration in multi-label problems. The method described in [a] introduces a new asymmetric loss function for calibration. In contrast, our method is a post-hoc method which can be applied after the training process and does not need access to the internal structure of the model. This post-hoc nature allows our method to be easily applied to any pre-trained model, providing additional flexibility and ease of use. In addition to this our method can be applied in conjunction with the approach described in [a]. After initial calibration by using the loss function described in [a] our method can further improve calibration. In the revised version, we will provide results by first training the model following the approach in [a] and then further improving the results using our post-hoc method.
>
>        [a]Cheng, Jiacheng, and Nuno Vasconcelos. "Towards Calibrated Multi-label Deep Neural Networks." Proceedings of the IEEE/CVF Conference on Computer Vision and Pattern Recognition. 2024.
>
>
> 4. We appreciate the importance of conducting comprehensive experiments across a wide range of models to demonstrate the generalizability of our method. In our current work, we selected ResNET50 as it is the most commonly used model in the multi-label classification task and to provide a strong initial demonstration of effectiveness of our method. The results we obtained confirm the benefits of our approach, showing improved calibration performance on the model. However we recognise the importance of including ViT in our experiments and will include it in the revised version.
>
>       Varying the number of bins and choosing the best number among that is a different kind of work which is discussed in [b]. We need to calculate the bayesian score of each binning configuration to get the best binning settings. In future we will extend our work further to study this.
>
>               [b] Naeini, Mahdi Pakdaman, Gregory Cooper, and Milos Hauskrecht. "Obtaining well calibrated probabilities using bayesian binning." Proceedings of the AAAI conference on artificial intelligence. Vol. 29. No. 1. 2015.

---

### Review · Reviewer_QaRG · 2024-08-29

**Summary Of Contributions:**

This paper proposes a theoretical framework to calibrate multi-label classification problems, and verifies the theoretical results by experimental evaluation on multiple datasets and models.

**Audience:**

Yes

**Claims And Evidence:**

No

**Requested Changes:**

Please see the "Strengths And Weaknesses" section!

**Strengths And Weaknesses:**

**Major concerns**

1. This paper is poorly written, which sometimes makes it diffuclt to follow.

2. The authors first introduce two metrics, ECE_total and ECE_wavg, and then employ their own metrics to show that their method for calibration outperforms the uncalibrated scenario!! How do we know the introduced metrics are suitable ones at the first place?

3. The proposed framework for calibration of multi-label classification problems is simply applying the single-label version independently to each label, assuming that the labels are independent of each other. Is that a realistic assumption for multi-label scenarios in practice?

4. The authors used "hamming loss" as a metric to evaluate the accuracy of the calibrated and uncalibrated methods. I think accuracy or AuRoC metrics per label are more informative here regarding the performance of the methods.


**Minor concerns**

1. In Theorem 1, differentiation is performed with respect to which parameter(s)?

2. The motivation behind calibrated models are not emphasized much in the introduction. Why do we need more calibrated models?

---

> ### Author Response · Authors · 2024-09-07
>
> Thank you for your thoughtful suggestions.
>
>         The responses are provided in the same order as the questions for clarity
> $\textbf{Answers to Major Concerns}:$
> 1. Thank you for your feedback regarding the clarity of our writing. We apologise for any confusion caused by the presentation of our work. To address this, we will review the manuscript again and improve its readability and coherence.
>
> 2. The metrics we introduced, ECE_total (Expected Calibration Error Total) and ECE_wavg (Weighted Average Expected Calibration Error), are specifically designed to address the unique aspects of multi-label problems. These two metrics are based on the original ECE metric proposed by [a]. In ECE_tot we calculated the total ECE of all labels. ECE_wavg is adapted to better suit multi-label settings by weighting the calibration error of each label by the fraction of labels present. This is to deal with scarcity of a particular label in a dataset which is very common in multi-label problems. Generally it increases the ECE of that label because of poor estimation. Our metrics are tailored to account for this situation in case of multi-label problems.
>
>        [a]Naeini, Mahdi Pakdaman, Gregory Cooper, and Milos Hauskrecht. "Obtaining well calibrated probabilities using bayesian binning." Proceedings of the AAAI conference on artificial intelligence. Vol. 29. No. 1. 2015.
>
> 3. In our current framework, we assume label independence for simplicity and tractability. This assumption allows us to extend existing single-label calibration techniques to the multi-label setting in a straightforward manner. Furthermore, we provide a rigorous theoretical justification of the calibration method for both binary and multi-label settings which demonstrates the efficacy of the extension with the assumption. Also assuming independence is particularly relevant in scenarios where the labels are weakly dependent.
>
>    However, we acknowledge that labels are often dependent in practical multi-label scenarios, which can impact calibration performance. We plan to address this in future work by incorporating techniques that model label dependencies, such as joint calibration methods or dependency-aware loss functions, to improve performance in more complex settings.
>
> 4. We appreciate your suggestion to consider accuracy or AuRoC metrics per label, as they can indeed provide additional insights into the performance of calibration methods. We chose Hamming loss as our primary evaluation metric because it directly measures the number of misclassified labels, which is particularly relevant in multi-label classification problems where each instance can have multiple labels. Hamming loss provides a straightforward and interpretable measure of overall performance by accounting for both false positives and false negatives across all labels. This metric is especially useful in scenarios where all labels are equally important, which aligns with the goals of our calibration method.
>           While Hamming loss effectively captures the aggregate performance across all labels, we recognize that per-label AuRoC metrics are also important as it can offer more detailed insights into the performance of individual labels. To address your suggestion, we are open to including additional analyses using AuRoC metrics per label. This would complement the Hamming loss results and provide a more comprehensive evaluation of our method's performance. We believe that incorporating these metrics will strengthen our study and offer a more complete picture of how well our calibration method works in different scenarios.
>
> $\textbf{Answers to Minor Concerns}: $
> 1. We apologise for not mentioning the variable respect to which we differentiate. The differentiation is done with respect to $\theta_m$.
>
> 2. We agree that clearly emphasising the motivation for calibrated models is essential to frame the significance of our work. Calibration is crucial in machine learning, particularly for models which predict probability, because it directly affects the reliability of predicted probabilities. Well-calibrated models ensure that the predicted probabilities correspond closely to the actual likelihoods of events, which is vital in many applications, such as, medical imaging, disease detection, autonomous car driving. For example, in healthcare, calibrated models can provide more accurate risk predictions, leading to better patient outcomes. In multi-label scenarios, where multiple decisions are made simultaneously, the need for accurate and reliable probability estimates becomes even more critical. In response to your suggestion, we will revise the introduction to better highlight these motivations. We will emphasise the practical implications of calibration, such as improving decision-making accuracy and reducing uncertainty in predictions, which are essential in fields like healthcare, finance, and autonomous systems.

---

### Decision · Action_Editor_konz · 2024-10-27

**Recommendation:** Reject

**Comment:**

In this paper, the authors establish a theoretical basis for why histogram binning consistently enhances the calibration and accuracy of neural network outputs, extending this approach to the multi-label setting. They provide some experimental evaluation across multiple datasets.

The main concern is that the theoretical claims are not sufficiently backed by either theoretical or empirical evidence. For instance, the authors claim that their calibration method outperforms the uncalibrated scenario; however, as Reviewer QaRG noted, this claim is supported primarily by metrics introduced by the authors, raising questions about their suitability. Additionally, the calibration approach assumes label independence in multi-label classification, which is often unrealistic in practice. Reviewer 82Zt further highlighted that the experiments do not sufficiently support the claims and suggested an in-depth ablation study. Reviewers also noted that the paper could benefit from improved clarity and organization.

After the rebuttal, all three reviewers are NOT in favor of accepting the paper. Following their recommendation, I am unable to support its acceptance.

**Audience:**

Might be of interest to TMLR  audience.

**Claims And Evidence:**

The convincing evidence does not quite support the paper's claim. See the details below.